# Life Satisfaction among Self-Employed People in Different Welfare Regimes during the COVID-19 Pandemic: Significance of Household Finances and Concerns about Work

**DOI:** 10.3390/ijerph20065141

**Published:** 2023-03-15

**Authors:** Mikael Nordenmark, Bodil J. Landstad, Åsa Tjulin, Stig Vinberg

**Affiliations:** 1Department of Health Sciences, Mid Sweden University, 831 25 Ostersund, Sweden; 2Unit of Research, Education and Development, Ostersund Hospital, 831 83 Ostersund, Sweden; 3Faculty of Human Sciences, Mid Sweden University, 831 25 Ostersund, Sweden

**Keywords:** COVID-19, life satisfaction, self-employed, welfare regimes

## Abstract

Most studies have shown that self-employed people have higher levels of life satisfaction than employed people due to high levels of job satisfaction, work flexibility, and job control. However, during the COVID-19 pandemic, many self-employed people experienced economic strain and worried about the situation of their employees and survival of the company. The aim of this study was to examine the level of life satisfaction among self-employed people during the COVID-19 pandemic in different welfare regimes. Analyses were mainly based on Eurofound’s “Living, Working and COVID-19” online survey. Fieldwork took place between April and June 2020 in 27 EU countries. The results showed that there was a significantly lower level of life satisfaction among self-employed people than employed people during the pandemic. This was in contrast to analyses from approximately one year before the pandemic, which showed that life satisfaction was higher among self-employed people. The main reasons for this lower level of life satisfaction among the self-employed during the pandemic were worse household financial situation and more worries about their job. Analyses of life satisfaction among the self-employed in different welfare regimes indicated that self-employed people in the Nordic welfare state regime largely kept their relatively high level of life satisfaction during the pandemic, but this was not the case for self-employed people in other welfare regimes.

## 1. Introduction

In 2018, self-employment in European countries encompassed a large number of individuals, with 32.6 million people aged 15 to 74 considered to be self-employed, representing 14% of the total EU employment [1]. Most self-employed people are either sole proprietors or have micro-sized (up to 10 employees) or small (up to 50 employees) businesses. The proportion of self-employed people who do not employ others has increased, as has the proportion of self-employed people in the service and public sector, while there has been a decline in the agriculture sector [2]. Government actors and policymakers want to stimulate self-employment because they believe it is a way to boost innovation, job creation, and economic growth [1,2].

At an individual level, many people choose self-employment for reasons such as gaining more autonomy and control of work and self-management of their working lives compared to organizationally employed people [2]. However, extensive research has shown that self-employed people are exposed to higher levels of uncertainty and responsibility, more intense time pressures, and longer working hours [3]. These positive and negative aspects can lead to different health-related outcomes for different groups of self-employed people. For example, while many European self-employed individuals report high job quality and well-being, around one-fifth report being self-employed out of necessity with little autonomy, a worse level of job quality, and lower well-being [2]. It can be assumed that the COVID-19 pandemic has affected many self-employed people through increased work demands, economic difficulties, uncertainty, and worries about the future, which may negatively affect health-related outcomes, such as life satisfaction. 

The pandemic and the subsequent restrictions applied by governments have put businesses under severe strain across the world [4]. Because the self-employed are more vulnerable with limited in-house resources compared to large businesses, it can be assumed that they have been more negatively affected by the pandemic [5,6]. Studies during the first phase of the pandemic showed that European self-employed people reported a significantly worse household financial situation and higher job insecurity than organizational employees [7]. According to a recent investigation from Eurofound [8], in spring 2021, around half of self-employed people reported having a financially fragile situation, meaning that without an income, their savings would not be enough to maintain their usual standard of living. Another result indicated that 46% of self-employed people reported difficulties in making ends meet. In addition, the reduction in working hours and loss of income was found to contribute to reduced subjective well-being for self-employed people compared to employed individuals [9]. However, for some self-employed people, the pandemic has been advantageous for their business, bringing greater profitability due to increased demands for their products and services [10]. 

It is essential to explore how the self-employed and their enterprises have been influenced by the pandemic because they are seen as key drivers of economic development and constitute an increasingly prominent group in many countries [2]. In addition, research has shown a strong relationship between financial hardship and impaired well-being among the self-employed [11]. In many countries, governmental support, such as income protection, paid sick leave, adjustment support, and financial support, has been targeted at the self-employed [4]. Although these support measures are comprehensive, a large number of self-employed people do not apply for financial support because they do not fulfill the requirements for applications or because they are not sure whether they are eligible [10,12]. In addition, in spring 2021, self-employed people in Europe reported a significantly lower level of trust in governmental institutions compared to results from the corresponding period in 2020 [8]. 

### 1.1. Life Satisfaction among Self-Employed People

In earlier research, being a self-employed individual has been characterized as a stressful job, with lower average earnings compared to employees [3]. Working for oneself entails high levels of stressors, such as uncertainty, high workload, long working hours, and complex tasks [3,13,14]. On the other hand, the job brings a high degree of autonomy with the freedom to decide what and how to perform different work tasks [3,13,15]. Theories describe self-employment as healthy and “active jobs“ [3,16] with a combination of high job demands and high job control. However, it is important to consider the differences between various groups of self-employed people, with some being attracted to self-employment while others are self-employed out of necessity [2]. These differences are supported by a study of European self-employed people, which identified distinct profiles among different groups of self-employed individuals (unhappy, languishing, happy, satisfied, passionate, and flourishing) and substantial differences in work-related variables and well-being [17]. 

Concerning health-related outcomes, studies have shown mixed and inconclusive results [6]. Many studies have shown that self-employed people are healthier, happier, and more satisfied with their job than employees [15,18,19]. There are also studies that show better physical health among the self-employed. Stephan and Roesler [15], for example, found that self-employed people had better physical health, lower blood pressure, and fewer somatoform disorders compared to employees in a German national representative study. In a study from the United States [20], self-employment was positively associated with perceived physical health and negatively associated with having diabetes, high blood pressure, high cholesterol, and arthritis. A Swedish study using register data [21] showed that mortality was lower among self-employed people who ran limited-liability companies but higher among those operating as sole proprietors compared to employed people. Over and above the fact that self-employed people have better health-related outcomes than employees in the majority of studies, they also have high levels of job control and flexibility and a strong feeling of pursuing their goals [22]. 

Some studies have shown negative associations between self-employment and health outcomes. Jamal [23] found that the self-employed experienced more psychosomatic health problems compared to employed individuals in Canada. Benavides et al. [24] studied self-employed individuals in EU countries and found that they were more likely to report fatigue compared to employees. Lee and Kim [25] found that people who chose self-employment in Korea experienced significantly higher odds for subjective poor health. There are also studies showing no differences in health-related outcomes between self-employed and employed people [13,26,27]. 

When looking at subjective measures of health-related outcomes, different terms are used, such as mental health, well-being, and life satisfaction. Life satisfaction is a general indicator that is positively related to satisfaction with aspects of work, family, and leisure time [28]. For self-employed people, job satisfaction is more closely associated with everyday life satisfaction compared to employees, reflecting the centrality of work in their life [29]. Research into life satisfaction among the self-employed has shown varied results, as have studies on several other health-related outcomes. For example, Schjoedt and Shaver [30] found no evidence of increased life satisfaction for the self-employed in the US, while van der Zwan et al. [31] found no differences in life satisfaction between self-employed and employed people in the UK. Some studies have also indicated negative impacts when individuals become self-employed [29,32]. In contrast to these results, several studies have shown positive results related to life satisfaction for the self-employed. Blanchflower and Oswald [33] and Prottas and Thompson [34] found positive relationships between self-employment and life satisfaction in a US sample, and similar results were found in a German sample [15]. 

Researchers have tried to explain these contradictory results through differences in working conditions, motivation for becoming self-employed, and other factors [35]. For example, work characteristics such as high autonomy and job control have positive effects on life satisfaction [22]. Regarding motivational factors, research in several countries has shown that individuals who leave employment and are attracted by entrepreneurial opportunities have high life satisfaction [3]. Decreased life satisfaction has been found in individuals who are pushed into self-employment from unemployment [19,36]. Other factors that are associated with life satisfaction include business success (or lack of success) [11,37], type of self-employment [38], personality traits, and cultural differences in the entrepreneurial climate of different countries [39]. In addition, research has shown that self-employed people who have employees seem to report higher life satisfaction than both employees and sole proprietors [19,40]. 

Common explanations behind self-employed people often reporting better results for both objective and subjective health outcomes, including life satisfaction, are that they have high levels of job control and a strong feeling of pursuing their goals [22]. Another explanation, recently discussed by Stephan et al. [6], is something referred to as self-selection. Individuals who are healthy and stress-resistant may choose self-employment more often than other individuals [3]. On the other hand, individuals with poor health may be attracted to self-employment as it can offer the flexibility to decide on one’s work and therefore help to deal with physical or mental health issues [41].

### 1.2. Life Satisfaction among Self-Employed People in Different Welfare Regimes

A common explanation referred to when discussing variations in well-being and life satisfaction is the welfare state regime. Welfare state regimes are seen as major determinants of differences in health, well-being, and life satisfaction as they can moderate the effects of socioeconomic conditions on these outcomes. Life satisfaction has become an important indicator when measuring the quality of life in different countries and welfare regimes, complementing other indicators such as GDP and life expectancy [42,43]. However, it is likely that no single aspect of welfare regimes is responsible for well-being and life satisfaction, with its influence instead connected to a combination of policies [44].

In Europe, there are main differences between different welfare regimes in terms of financial and social support from various governments to employed and self-employed people experiencing financial hardship. This means that differences may exist in how people in different welfare regimes have managed to adapt to the pandemic and maintain a high level of life satisfaction. In the literature relating to classification of different social welfare regimes, five different types of welfare regimes have been identified in Europe: the Nordic (or social democratic) model, the conservative (or corporatist) model, the liberal model, the Southern (or Mediterranean) model, and the Eastern European (or post-socialist) model [45,46,47,48]. The data used in this study does not include any country representing the liberal model, so the brief description of the different welfare regimes below does not include the liberal welfare regime.

The Nordic welfare regime includes the Nordic countries and is often described as a well-developed policy context that offers universal social rights and decommodification of social rights. Its attributes are encouragement of independence in combination with general social policy schemes. Among the countries situated in other parts of Europe, there is great heterogeneity regarding the types of welfare state arrangements. Countries located in Southern and Central Europe are most commonly categorized in one of two clusters: conservative and Southern European. However, social policy in these countries is generally characterized as relatively passive and based on social contributions rather than taxes, moderate redistribution of income, and values such as restriction of market-distributed welfare and conservation of traditional family forms and norms [45,46,47,48]. Former Eastern European countries have not really found their place in the research literature on welfare regimes and are difficult to classify as a distinct welfare regime type. Former Eastern European countries have been characterized by an extensive social policy that encourages full-time employment for both men and women, but the development of social protection systems over the past 20 years varies considerably between countries [45,48]. 

In general, studies have found that there are smaller inequalities in well-being and life satisfaction in the Nordic welfare regime compared to the other regime types. A major explanation for this is that the Nordic welfare regime does more to reduce financial stress for people in economic hardship [49]. Relatively few studies have specifically analyzed what the different welfare regimes mean for the self-employed experiencing economic hardship and difficulties with their businesses. A study by Spasova et al. [48] analyzed how the social protection systems in European countries influenced social risks for the self-employed. The results indicated that self-employed and employed people benefit similarly from the universal social rights provided by the Nordic welfare regime. In other welfare regimes, the difference in access to social protection was greater between self-employed and employed people and between different categories of self-employed people. Consequently, in times of economic hardship, the self-employed may be better off in the Nordic welfare regime compared to other welfare regimes. 

### 1.3. Research Problem, Aim, and Hypotheses 

Most of the earlier research concerning life satisfaction among the self-employed has shown that they have higher life satisfaction than employed people. However, there are reasons to believe that self-employed people in particular have been negatively affected by the COVID-19 pandemic because of increased concerns about their business and the situation of their employees as well as increased economic difficulties. It is therefore important to investigate the life satisfaction of self-employed people during the pandemic as well as the explanations for the potential decrease in life satisfaction. It is also interesting to study whether the level of life satisfaction varied between self-employed people according to differences in welfare state arrangements during economic crises. 

Therefore, the aim of this study was to analyze the level of life satisfaction among self-employed people in different welfare regimes during the COVID-19 pandemic. The following hypotheses were analyzed:There was a lower level of life satisfaction among the self-employed than employed people during the COVID-19 pandemic.The level of life satisfaction among self-employed people was lower during the pandemic than before compared to employed people.The potential lower level of life satisfaction among the self-employed during the pandemic can mainly be explained by economic difficulties and greater concerns about their work.During the pandemic, the life satisfaction among the self-employed was higher in a more extensive welfare regime, such as the Nordic welfare regime, than in the Eastern European, conservative, and Southern European welfare regimes.

## 2. Materials and Methods

### 2.1. Data 

This study was based on a large-scale online survey that took place during the first phase of the COVID-19 pandemic between 9 April 2020 and 11 June 2020 in 27 EU countries [7]. Data were obtained from Eurofound (https://www.eurofound.europa.eu/data/covid-19 (accessed on 6 March 2023) upon request from the authors. Entitled “Living, Working and COVID-19”, the aim was to investigate the impact of the pandemic on well-being, work, and the financial situation of individuals across the EU. The survey consisted of about 35,500 respondents, including 30,000 employed individuals and 5500 self-employed people, from 27 European countries in different parts of Europe. The survey applied a nonprobability sampling method, but data were weighted according to different background characteristics to ensure that they would be fairly representative of all the included countries. Most of the questions in this study were based on Eurofound’s European Quality of Life Survey (EQLS) and European Working Conditions Survey (EWCS), while some questions were new. Data from the European Social Survey (ESS) in 2018 was used as an indicator of levels of life satisfaction among employed and self-employed people before the pandemic. The ESS included the same question on life satisfaction and included similar countries to those in the “Living, Working and COVID-19” survey.

### 2.2. Variables

The dependent variable “life satisfaction” was indicated by the question, “All things considered, how satisfied are you with your life these days?” Answers varied on a scale from 1 = very dissatisfied to 10 = very satisfied. The independent variable “employment status” was measured by the question, “Which of these categories best describes your situation” (1 = self-employed, 0 = employed). Two independent variables measured concerns about work. “Job insecurity” was measured by the question, “Using this scale, how likely or unlikely do you think it is that you might lose your job in the next 3 months” (1 = very unlikely, 5 = very likely), while “concerns about work” was measured by the question, “How often in the last two weeks have you been worried about work when you were not working” (1 = never, 5 = always). Two variables were used for measuring household finances. “Poor household finances” was measured by the question, “Thinking of your household’s total monthly income, is your household able to make ends meet?” (1 = very easily, 6 = with great difficulty), while “deteriorating financial situation” was measured by the question, “When you compare the financial situation of your household three months ago and now would you say it has become better, worse or remained the same” (1 = better, 3 = worse).

Welfare state regime was measured by countries representing different regimes and was based on earlier research on welfare regimes [45,46,47]. The Nordic welfare regime consists of Denmark, Finland, and Sweden (N = 1870). The Eastern regime cluster consists of Bulgaria, Croatia, the Czech Republic, Estonia, Hungary, Latvia, Lithuania, Poland, Romania, Slovakia, and Slovenia (N = 8662). The conservative welfare regime comprises Austria, Belgium, France, Germany, Ireland, Luxemburg, and Netherlands (N = 15,931). The Southern regime consists of Cyprus, Greece, Italy, Malta, Portugal, and Spain (N = 10,078).

The multivariate analyses controlled for a number of background characteristics: gender (1 = woman, 0 = man), age (18–80 years), cohabiting/married (1 = cohabiting/married, 0 = single), and children (1 = children living at home, 0 = no children living at home).

### 2.3. Analysis Strategy

The analyses were structured in the following way. First, the distribution of dependent and independent variables across employed and self-employed people in Europe was analyzed. The next step involved analyzing life satisfaction among self-employed people in relation to employed people using an OLS regression in four different models that controlled for other independent variables at different steps. This was followed by an illustration of the bivariate values for dependent and main independent variables in the various welfare regimes by reporting mean and Eta values. The final part of the empirical section was an analysis of life satisfaction among self-employed people in relation to employed people using separate OLS regression models for the various welfare state regimes.

## 3. Results

The results from ESS 2018, presented in Table 1, show that there was a somewhat, but significant, higher level of life satisfaction among self-employed people than among employed people in Europe before the pandemic. During the pandemic, the levels of life satisfaction were lower among both self-employed and employed people than before. However, the decrease in life satisfaction was greater among self-employed people than employed people. Consequently, there was a significantly lower level of life satisfaction among self-employed people (6.10) than among employed people (6.45) during the pandemic, which supports hypotheses 1 and 2. There was no significant difference between self-employed people who had employees compared to those who did not (6.15 verses 6.08). 

Self-employed people experienced poorer household finances and more concerns about their job than employed people during the pandemic. About 32% of self-employed and 20% of employed people reported their household finances as being poor. As much as 64% of self-employed people reported that their financial situation had become worse during the last three months compared to 35% of employed people. Further, 50% of self-employed people worried about their work compared to 27% of employed people. Finally, 22% of self-employed people experienced job insecurity during the pandemic compared to 13% of employed people.

Table 1 shows that there are some significant differences between self-employed and employed people in terms of different background characteristics. A lower percentage of self-employed people were women (40%) compared to women in employment (49%). Self-employed people were generally older at around 48 years compared to 44 years for employed people. It was more common for self-employed people to be cohabitants or married compared to employed people, and it was somewhat less common for self-employed people to have children living at home.

In agreement with the results in Table 1, Model 1 in Table 2 shows that self-employed people in Europe had a significantly lower level of life satisfaction than employed people during the pandemic. However, when controlling for household finances and concerns about their job, the level of life satisfaction rose for the self-employed. When controlling for variables indicating the economic situation in the household in Model 2, the life satisfaction among the self-employed became significantly higher (*p* = 0.1). Further, when also including measures of the level of concern about their job (Model 3), the level of life satisfaction became significantly higher among self-employed compared to employed people. Even when controlling for different background characteristics, as seen in Model 4, the level of life satisfaction was significantly higher among the self-employed. This indicated that the main reasons for the lower level of life satisfaction among self-employed people in Europe during the pandemic were worse household financial situation and greater concerns about their job compared to employed people. This result supports hypothesis 3.

Poor household finances and a deteriorating financial situation were related to significantly lower levels of life satisfaction in all models, but the relationships were somewhat weakened when controlling for concerns about work in Model 3. Worries about work and job insecurity also significantly increased the risk of lower life satisfaction, as shown in Models 3 and 4. Background characteristics were associated with life satisfaction in the following way: women had a significantly lower level of life satisfaction than men, and the level of life satisfaction increased with age; cohabiting/married people had a significantly higher level of life satisfaction than single people; and people with children living at home were more satisfied than people with no children living at home.

Table 3 analyzes the main independent variables related to life satisfaction among employed and self-employed people in different welfare regimes by illustrating the mean and Eta values. There was a somewhat higher level of life satisfaction among self-employed people in the Nordic welfare regime but significantly lower levels of life satisfaction in the Eastern, conservative, and especially Southern welfare regimes during the pandemic. 

Results regarding household finances and concerns about work in general (apart from job insecurity in the Nordic welfare regime) showed that self-employed people experienced significantly poorer household finances and more concerns about their job in all types of welfare regimes. However, in most cases, the differences between self-employed and employed people were greater in the Eastern, conservative, and Southern welfare regimes than in the Nordic welfare regime. This meant that self-employed people in these regimes generally experienced greater problems related to their household finances and more concerns about their job than self-employed people in the Nordic welfare regime. These results indicated that lower levels of life satisfaction among the self-employed in the Eastern, conservative, and Southern welfare regimes might, to some extent, be explained by poorer household finances and more concerns about their job. This is further analyzed in Table 4.

Table 4 presents separate OLS regressions for the different welfare regimes and analyzes the relationship between employment status and the level of life satisfaction when controlling for household finances, concerns about work, and background characteristics during the pandemic. Model 1 shows the bivariate relationships between employment status and life satisfaction in the different welfare regimes. There was almost a significantly higher level of life satisfaction among self-employed people in the Nordic welfare state regime. There were significantly lower levels of life satisfaction among the self-employed in all other welfare state regimes, and this was particularly obvious in the Southern welfare state regime. This indicates that self-employed people in the Nordic welfare state regime largely retained their relatively high level of life satisfaction, but this was not true for the self-employed in other welfare regimes, especially in the Southern welfare state regime, which supports hypothesis 4.

Model 2 shows the relationship between employment status and life satisfaction in different welfare regimes when controlling for household finances, concerns about work, and background variables. When controlling for these variables, the significantly negative relationship between self-employment and life satisfaction disappeared in the Eastern and Southern welfare regimes and became positive in the conservative welfare regime. These results indicated that the lower levels of life satisfaction of self-employed people in the Eastern, conservative, and Southern welfare regimes could, to some extent, be explained by poorer household finances and more concerns about their job. The level of life satisfaction among self-employed people in the Nordic welfare regime remained at the same level when controlling for the other variables. The R^2^ values were generally quite low, indicating that the level of life satisfaction could, to a large part, be explained by other factors not included in this study. Other possible factors of importance would include, for instance, socioeconomic status, health status, social relationships, and personal characteristics. 

## 4. Discussion

To our knowledge, there are no studies analyzing life satisfaction among self-employed people in relation to employees and different welfare regimes during the COVID-19 pandemic, which has been carried out in this study. The results showed that the level of life satisfaction was significantly higher among self-employed people compared to employees before the pandemic but significantly lower among the self-employed during the pandemic. In addition, self-employed people reported poorer household finances and a higher degree of concern about their job. Regression analyses indicated that the main reasons for the lower life satisfaction among the self-employed during the pandemic were poorer household financial situation and increased concerns about their job. Results concerning different welfare regimes showed a higher level of life satisfaction among the self-employed in the Nordic welfare regime but significantly lower levels of life satisfaction in the Eastern, conservative, and especially Southern welfare regimes. The results indicated that lower levels of life satisfaction among the self-employed in the Eastern, conservative, and Southern welfare regimes could, to some extent, be explained by poorer household finances and more concerns about their job. These findings highlight the need for support from a range of organizations, including governmental authorities, business organizations, and occupational health consultants, for self-employed people during and after the pandemic. 

A somewhat surprising result is that the self-employed experienced lower levels of life satisfaction during the pandemic compared to employees. Most studies have shown that self-employed people report a higher level of subjective health, physical health, and job satisfaction than employees [3,21,35]. The same pattern can be found for life satisfaction, although research indicates mixed results concerning this health outcome [3,35]. Explanations for these results may be that the pandemic has contributed to greater economic uncertainty and concerns about how the demand for their services will evolve. Recent studies have shown that the COVID-19 pandemic has had negative consequences on many self-employed people, generating increased strain, a worse household situation, and loss of customers and income [7,8]. In addition, studies report that many self-employed people have had problems accessing the different support measures available from governmental authorities [10,12]. Our regression results confirmed that a worse household financial situation and concerns about work (worries about work and job insecurity) were explanatory variables for a lower level of life satisfaction among self-employed people during the pandemic. 

Similar to earlier studies on the significance of different welfare regime types in relation to people’s living conditions and well-being [49], the results from this study indicated that the level of life satisfaction seemed to be higher in the Nordic welfare regime. Interestingly, self-employed people appeared to benefit even more than employed people from living in a Nordic welfare regime context during the pandemic. A possible explanation could be that self-employed people were particularly negatively affected by the pandemic due to greater concerns about their business and the situation of their employees as well as major economic difficulties. In such a situation, it would be especially beneficial to live in a welfare regime context that offers comprehensive financial and social support during economic hardship. This possible explanation is in some sense supported by the results of the separate regression analyses of different welfare regimes. These showed that the lower levels of life satisfaction among self-employed people in the conservative, Southern, and Eastern welfare regimes disappeared when controlling for variables measuring the financial situation and worries about work.

### Strengths and Limitations 

Results generated from comparative statistics should be interpreted with some caution. First, the respondents in the survey were not recruited through a probability sampling method. They were recruited through snowball sampling methods and social media advertisements, which means that the sample is not representative. However, the sample was weighted on the basis of gender, age, education, and self-defined urbanization levels. This means that the data is representative of the population and fairly nationally representative [50].

Second, questions may be understood and interpreted differently in different cultural and national contexts. For instance, different societal traditions and cultures might possibly influence how the individual reports attitudes and level of life satisfaction, entailing a risk that the answers will reflect differences in how people respond to surveys rather than indicating genuine differences in attitudes [51]. 

Obviously, establishing the causal directions between policy and measures of financial and job situation and a subjective measure such as life satisfaction is difficult, especially in cross-sectional studies such as this, and it is reasonable to assume that these factors influence each other in complex ways. However, one can at least argue that most policies have been established before the individual is born, implying that at least some part of a relationship between policy and attitudes can be interpreted as an influence of policy on attitudes [52]. 

## 5. Conclusions

In conclusion, the results showed that there was a significantly lower level of life satisfaction among self-employed people compared to employed people during the pandemic. The main explanations for the lower level of life satisfaction among the self-employed during the pandemic were poorer household finances and more concerns about their job. Analyses of life satisfaction among the self-employed in different welfare regimes indicated that self-employed people in the Nordic welfare state regime had largely retained their relatively high level of life satisfaction during the pandemic, but this was not the case for the self-employed in other welfare regimes. The results indicated that the lower levels of life satisfaction among self-employed people in the Eastern, conservative, and Southern welfare regimes could, to some extent, be explained by poorer household finances and more concerns about their job.

These findings highlight the need for support from a range of organizations, including governmental authorities, business organizations, and occupational health consultants for self-employed people during and after the pandemic. One aspect that appears to be of major importance is financial support for the self-employed during economic crises to reduce worries about the survival of the company and the situation for their employees as well as their own financial situation. The results from the analyses of different welfare regimes highlight that comprehensive and universal social and economic support from governments is necessary for the self-employed during financial crises, such as a pandemic. 

Earlier research has shown that life satisfaction among self-employed people is affected by factors such as attraction to entrepreneurship, business success, personal traits, entrepreneurial climate, and number of employees. A theoretical implication from this study is that it is important to also consider situational factors, such as a pandemic crisis and cultural aspects of the country.

For future research, the relatively low R^2^ values in the regression analyses highlight the need to also consider other independent factors related to work and private life. In addition, longitudinal quantitative and qualitative interview studies can contribute to increased knowledge about mechanisms beyond the study results. 

## Figures and Tables

**Table 1 ijerph-20-05141-t001:** Life satisfaction, household finances, concerns about work, and background variables among employed and self-employed people (percentage in brackets and mean).

	Employed	Self-Employed
*Life satisfaction*		
Before the pandemic (ESS, 2018)	7.17	7.38 ***
During the pandemic	6.45	6.10 ***
*Household finances*		
Poor household finances (values 5–6)	(20) 3.20	(32) 3.79 ***
Deteriorating financial situation (value 3)	(35) 2.29	(64) 2.61 ***
*Concerns about work*		
Worries about work (values 4–5)	(27) 2.70	(50) 3.37 ***
Job insecurity (values 4–5)	(13) 2.03	(22) 2.57 ***
*Background variables*		
Woman	(49)	(40) ***
Age	43.47	47.88 ***
Cohabiting/married	(66)	(71) ***
Children	(40)	(38) *
N (ca)	30,000	5500

*** *p* = 0.001 * *p* = 0.05.

**Table 2 ijerph-20-05141-t002:** OLS regression. Life satisfaction among the self-employed when controlling for household finances, concerns about work, and background variables (B-coefficients).

	Model 1	Model 2	Model 3	Model 4
*Employment status*				
Self-employed	−0.343 ***	0.059 (*)	0.243 ***	0.191 ***
*Household finances*				
Poor household finances		−0.468 ***	−0.373 ***	−0.357 ***
Deteriorating financial situation		−0.458 ***	−0.264 ***	−0.299 ***
*Concerns about work*				
Worries about work			−0.273 ***	−0.276 ***
Job insecurity			−0.212 ***	−0.194 ***
*Background variables*				
Woman				−0.135 ***
Age				0.002 *
Cohabiting/married				0.365 ***
Children				0.068 **
R^2^	0.003	0.137	0.179	0.184

*** *p* = 0.001 ** *p* = 0.01 * *p* = 0.05 (*) *p* = 0.1.

**Table 3 ijerph-20-05141-t003:** Life satisfaction, household finances, and concerns about work among employed and self-employed people in different welfare regimes (mean and Eta values).

	Nordic Welfare Regime		Eastern Welfare Regime		Conservative Welfare Regime		Southern Welfare Regime	
	Empl	Self-Empl	Empl	Self-Empl	Empl	Self-Empl	Empl	Self-Empl
*Life satisfaction* Eta	7.21	7.52 (*) 0.043	6.25	6.13 * 0.021	6.61	6.41 *** 0.030	6.15	5.71 *** 0.082
*Household finances*								
Poor finances Eta	2.48	2.83 * 0.070	3.52	3.79 *** 0.076	2.99	3.62 *** 0.140	3.42	4.00 *** 0.168
Deteriorating finance sit. Eta	2.09	2.27 *** 0.083	2.40	2.60 *** 0.143	2.19	2.55 *** 0.202	2.42	2.55 *** 0.205
*Concerns about work*								
Worries about work Eta	2.48	2.63 * 0.046	2.63	3.12 *** 0.158	2.59	3.22 *** 0.171	3.01	3.75 *** 0.249
Job insecurity Eta	1.75	1.87 0.030	2.40	2.67 *** 0.086	1.82	2.18 *** 0.103	2.15	2.90 *** 0.227
N	1683	122	6783	1523	13,682	1820	7841	1974

*** *p* = 0.001 * *p* = 0.05 (*) *p* = 0.1.

**Table 4 ijerph-20-05141-t004:** OLS regression. Life satisfaction among self-employed people when controlling for household finances, concerns about work, and background variables in different welfare regimes (B-coefficients).

	Nordic Welfare Regime		Eastern Welfare Regime		Conservative Welfare Regime		Southern Welfare Regime	
	Model 1	Model 2	Model 1	Model 2	Model 1	Model 2	Model 1	Model 2
Self-employed	0.318 (*)	0.319 (*)	−0.125 *	0.043	−0.199 ***	0.462 ***	−0.446 ***	0.023
Poor household finances		−0.211 ***		−0.391 ***		−0.317 ***		−0.388 ***
Deteriorating financial situation		0.011		−0.334 ***		−0.405 ***		−0.113 *
Worries about work		−0.329 ***		−0.251 ***		−0.331 ***		−0.165 ***
Job insecurity		−0.120 *		−0.232 ***		−0.215 ***		−0.151 ***
Woman		−0.192 *		−0.093 (*)		−0.153 ***		−0.143 **
Age		−0.006		0.004 *		0.003 *		0.001
Cohabiting/married		0.597 ***		0.352 ***		0.381 ***		0.320 ***
Children		0.162 (*)		0.208 ***		−0.071 (*)		0.183 ***
R^2^	0.002	0.126	0.000	0.192	0.001	0.199	0.007	0.133

*** *p* = 0.001 ** *p* = 0.01 * *p* = 0.05 (*) *p* = 0.1.

## Data Availability

Restrictions apply to the availability of these data. Data were obtained from Eurofound [53] and are available from the authors with the permission of Eurofound.

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
