# Peer review of "Life Satisfaction among Self-Employed People in Different Welfare Regimes during the COVID-19 Pandemic: Significance of Household Finances and Concerns about Work"

_ijerph, 2023, doi:10.3390/ijerph20065141_

Round 1

Reviewer 1 Report

The authors present the results of a study on the differences in life satisfaction between employees and self-employed in different regions of the European Union, and analyze the effect of various individual factors as possible determinants of these differences, in the context before and during the pandemic. In my opinion, this is a very relevant study, with results of great value at both a psychosocial level and from which recommendations for rulers and politicians could be derived. I think there are some aspects that could improve the manuscript and I strongly encourage authors to take them into account:

1. Data. The authors mention that they make use of the European survey data. In my opinion, the authors need to indicate more clearly how they have obtained the data. Are these data freely accessible? Where is the data located? a web address?, etc.

2. Participants. Authors must include the number of participants or percentage of each regimen considered. How many participants come from Nordic welfare regime, Eastern, Conservative and Southern wel- 346 fare regimes?

3. Results. Table 1. Although statistical significance is observed in the comparison between employed and self-employed in life satisfaction, the difference is low (from 7.17 before the pandemic to 7.38, a difference of only 0.2 on a 10-point scale, or .30 after of the pandemic).

4. Line 283. "There is no significant difference between those self-employed people who have employees compared to those who do not (analysis not shown)". ¿Why don’t authors show that results?

5. Table 1. The authors have to explain how they categorized the groups for the presentation of the results on household finances and concerns about work. The data that appear in Table 1 are from those who responded 5/6 or those who responded more than 3?

6. Table 1. Why do they calculate the mean of Deteriorating financial situation when that is an ordinal variable with 3 points? This is not correct.

7. Table 1. I would like the authors to include percentages of all responses for household finances and concerns about work. How many participants answered 1, how many 2, how many 3…? Thus, the distribution of the responses of these variables could be better analysed.

8. The R2 are not high in any of the models, the authors should comment on this and offer some hypothesis of what other factors may be influencing.

Reviewer 2 Report

This paper is a comprehensive data review with 4 groups to be compared, plus 2 groups of employed and self-employed. It has significant data analysis and mixed findings. I found it was interesting as a novelty at first, however, I have to be careful reading the results and discussion as 4 groups are quite big and confusing to understand.

It is very useful if you could identify the strongest points for theoretical or practical contribution to be discussed and then focus on it. Not really sure as I think initially you aimed to compare 2 groups of employed and self-employeduring the pandemic and what is the importance of 4 welfare regimes for Europe?  discuss and highlight further.

Graphs or lines for comparing groups would also make your data presentation could be clearly depicted in figure form, in addition to your tables.

Reviewer 3 Report

Dear Authors, 

First of all, I thank you for the opportunity to review your article, which deals with a current topic and which raised interest in its reading.

The study is well structured, clearly written, and presents an extensive current literature review, which enriches the work in theoretical and empirical terms.

However, I only leave two suggestions for improvement, should they be considered relevant and applicable:

In point 2.2 - putting the information in tables or charts, makes the information more appealing, as they did in the next point of the results;

In point 4.1 - I would suggest referring to the proposal for future study, propose new methodologies, statistical analysis...

Overall I consider it a very enriching study.

Best regards
